# A Digital Workflow for the Fabrication of a Milled Removable Partial Denture

**DOI:** 10.3390/ijerph19148540

**Published:** 2022-07-13

**Authors:** Xing-Yu Piao, Jeongho Jeon, June-Sung Shim, Ji-Man Park

**Affiliations:** 1Department of Prosthodontics, College of Dentistry, Yonsei University, Seoul 03722, Korea; 18625049725@163.com (X.-Y.P.); jfshim@yuhs.ac (J.-S.S.); 2Central Dental Laboratory, College of Dentistry, Yonsei University, Seoul 03722, Korea; koreadenture@naver.com; 3Department of Prosthodontics & Dental Research Institute, Seoul National University, Seoul 03080, Korea

**Keywords:** CAD−CAM, digital manufacture, metal substructure, removable partial denture

## Abstract

Complete dentures fabricated with the additive or subtractive method have been widely used and proven to be clinically acceptable. However, fabrication of removable partial dentures (RPDs) using computer-aided design and computer-aided manufacturing is limited by its technique sensitivity as the pink resin, which encases part of metal framework, cannot be fabricated digitally. This article introduces a digital workflow to fabricate an RPD with the subtractive method. A complex structure of the offset metal framework and denture base with teeth sockets was milled with this technique. Artificial teeth were milled with a resin disk according to the computer-aided design data, resulting in the customized occlusal surface. This digital technique can be an alternative to the analog fabrication method as the RPD was fabricated digitally, keeping the original structures and reducing resin shrinkage on the intaglio surface.

## 1. Introduction

Removable dentures have been widely applied as a means of treatment to reconstruct oral function for edentulous patients [1,2,3]. Both the complete dentures and partial dentures place high demands on dentists and technicians. With the advances in the computer-aided design and computer-aided manufacturing (CAD−CAM) technique, fabrication of the complete dentures has been somewhat simplified and significantly improved [4,5,6].

Complete dentures, comprised of a resin base and artificial teeth, are frequently fabricated with the CAD−CAM technique via 3D printing or prefabricated resin disc milling [7]. Furthermore, such dentures are becoming more popular among dentists and showing better satisfaction by the patients when compared with the conventional dentures [8]. Additionally, metal frameworks have been manufactured with the additive or subtractive methods as well, showing a comparable or better fit than the conventionally cast fabricated ones [9,10]. However, fabricating a removable partial denture (RPD) using digital techniques is still a work in progress [11,12]. The current technique cannot be used to fabricate structures containing two different types of materials synchronously. To address this limitation, numerous studies have been developed: for example, bonding the denture bases, artificial teeth, connectors, and clasps digitally fabricated together; modifying the metal framework structure and modeling with the injection technique; and uniting the denture components via an additionally designed holding structure [12,13,14]. However, an excessive effort for the CAD or manual work is inevitable.

Hence, the objective of the present digital workflow for the fabrication of an RPD was to minimize the conventional analog working process while retaining regular denture components by milling a modified cast to hold the metal framework.

## 2. Materials and Methods

This technique consists of digitally designing the denture components and modifying the corresponding master cast data, followed by manufacturing the prosthesis with a milling machine.


Scan the master casts using a lab scanner (T500; Medit, Seoul, Korea) with a 7 µm scan accuracy (ISO 12836) and scan interocclusal relationship with the registered bite material. Export the data as the standard tessellation language (STL) format.Design the metal framework with a dental CAD software program (exocad DentalCAD; exocad GmbH, Darmstadt, Germany) after importing the casts’ data.Fabricate the metal framework via the lost-wax casting technique after milling a wax disk (Mazic Wax; VERICOM Co., Ltd., Chuncheon, Korea).Check the fit of the metal framework in the patient’s mouth and record the centric relation.Scan the centric relation record and assemble the digitized master casts accordingly.Design the denture base with teeth sockets and artificial teeth with a customized occlusal surface assembly with the CAD software program (exocad DentalCAD; exocad GmbH, Darmstadt, Germany). Export all data under the relative position to transfer numerical control data into the milling machine (Figure 1).Import the mandibular cast and RPD components data into a 3D modeling software program (Meshmixer; Autodesk Inc., San Rafael, CA, USA).Acquire the modified cast data with space for a base resin by selecting the region wider than the edentulous regions and intruding to 10 mm along the *Y*-axis with the aid of the “Extrude” function (Figure 2).Select the whole metal framework and offset by 0.5 mm to protect the milling bur from collision (Figure 3A). Acquire the complex of the denture base and offset metal framework by combining the two objects together on the software, and export the CAD data (Figure 3B).Mill the artificial teeth from a disk-type double cross-linked resin block (SR Vivodent CAD; Ivoclar Vivadent AG, Schaan, Liechtenstein). Mill the modified mandibular cast from the metal jig in which gypsum material was poured and set. The metal jig was attached to the milling unit (Rainbow Mill-Zr 2nd; Dentium Co., Ltd., Seoul, Korea).Seat the metal framework on the milled stone cast and pour autopolymerizing resin (Retec PRESS LT; Retec Kunststofftechnik GmbH, Rosbach, Germany) to fill the metal jig (Figure 4). Polymerize the resin with a pressure polymerization unit ((Palamat elite; Kulzer GmbH, Hanau, Germany) following the manual.Mill the denture base and offset metal structure complex after calculating the toolpath on a CAM software (hyperDent V8.2; FOLLOW ME! Technology GmbH, Munich, Germany) (Figure 5).Complete the RPD by removing offset resin and gypsum from the framework, bonding the milled teeth to the teeth sockets and polishing the prosthesis (Figure 6).


## 3. Discussion

The described workflow integrated a metal framework, artificial teeth, and a denture base manufactured via milling to fabricate the prosthesis. Milling the artificial teeth directly according to the design data permitted customization of the occlusal morphology. The fit of the RPD could be ensured by evaluating the framework intraorally ahead of time. Furthermore, the metal framework could be fabricated with other digital methods, such as metal printing or milling. To avoid shrinkage of the denture base that occurred during resin polymerization, a wider region for resin pouring was prepared by extruding the edentulous area. Hence, shrinkage at the intaglio surface of the denture base during conventional packing or injection methods can be avoided, resulting in better adaptation. In addition, the milled cast structure supported the metal framework in the right position during the milling process. With the accomplished design data stored, it is convenient to reproduce the prosthesis in case of any fracture.

As a functional impression is needed to fabricate this free-end RPD, master casts are poured and scanned in advance. However, for the cases requiring only an anatomic impression, a functional impression can be avoided by using an intraoral scanner as the new generation intraoral scanner has proved to be equipped with high accuracy in edentulous and partially edentulous arches [15]. Moreover, if a stable bite registration can be acquired with an intraoral scanner, the manual centric relation record in this study can also be avoided. Therefore, if the whole workflow can be simplified even further, prosthesis design can directly proceed taking intraoral scans at the patient’s first visit.

In a previous study, Lo Russo et al. provided a digital workflow to fabricate an RPD by using a polyetheretherketone framework instead of a metal framework [16]. Artificial teeth, flange, and framework were individually fabricated via the digital technique and bonded together. However, the conventional structure of the RPD was modified. The framework covered a larger area and contacted the tissue surface directly without resin surrounding it. Nishiyama et al. also suggested a similar method to make an RPD by substituting the framework material while the present study fabricated the RPD based on the regular design and maintained the common material [13].

The limitation of this technique is the requirement of a 0.5 mm offset design for the metal framework to protect the milling bur, which can be omitted with improvements in the milling machine. Teeth sockets should be carefully checked on the software after the metal framework is offset as the offset metal framework might protrude from the surface of teeth sockets if the space between them is too small. Furthermore, this digital workflow requires a high trueness for the metal framework fabricated with the subtractive or additive digital method, and the operator needs to remove the adherent resin and gypsum after milling. Additionally, due to the RPD module setting of software, connectors cannot be added between artificial teeth. Therefore, artificial teeth need to be bonded in the sockets separately, which may result in teeth displacement when compared with design data.

For further research, a clinical study can be prepared to evaluate the adaptation and teeth movement of the RPDs fabricated with this method, and more studies can be conducted to fabricate other dental prostheses, such as overdentures and metal-reinforced complete dentures, according to the method proposed, and evaluate the clinical feasibility.

## 4. Conclusions

This research introduced a digital workflow to fabricate an RPD by bonding artificial teeth to the denture base, which was milled along with a metal framework seated on the stone cast. The RPDs fabricated by the described method contain a customized occlusal morphology and less resin shrinkage on the base part. This approach provides a digital alternative to the conventional fabrication method and greatly reduces the manual operation.

## Figures and Tables

**Figure 1 ijerph-19-08540-f001:**
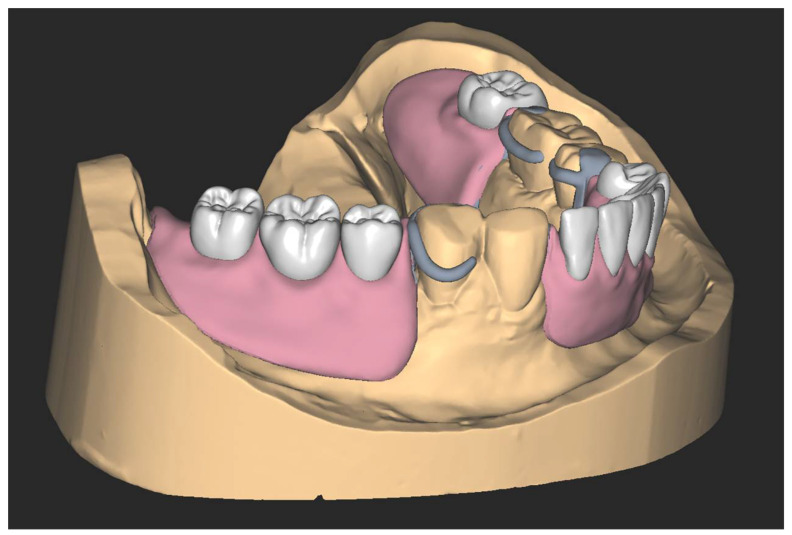
Virtual design of removable partial denture.

**Figure 2 ijerph-19-08540-f002:**
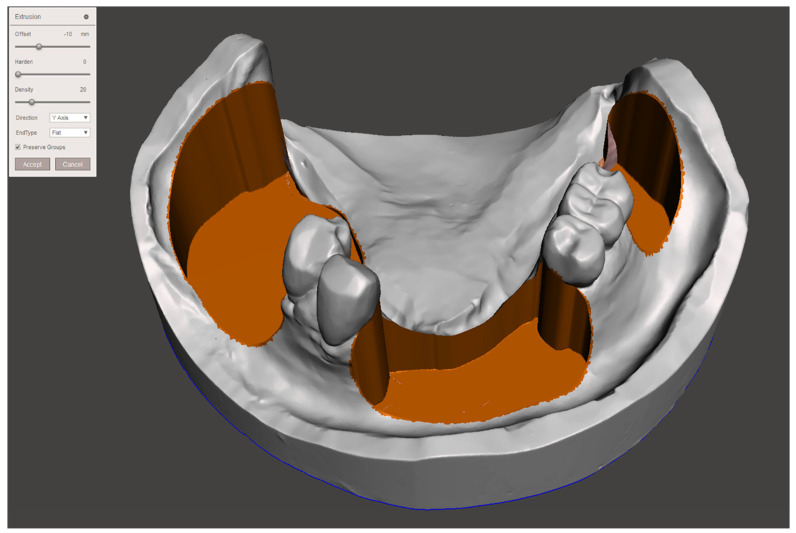
Virtual design of the modified mandibular cast.

**Figure 3 ijerph-19-08540-f003:**
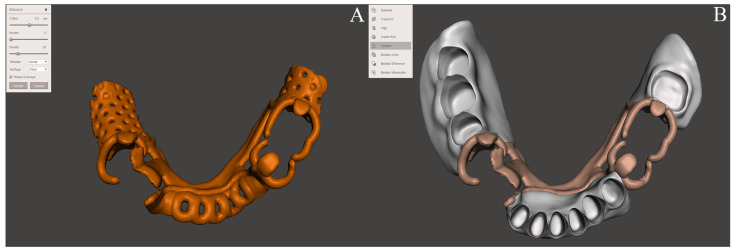
Virtual design of the denture base and offset metal framework complex. (**A**) Offsetting metal framework. (**B**) Combining the denture base and offset metal framework.

**Figure 4 ijerph-19-08540-f004:**
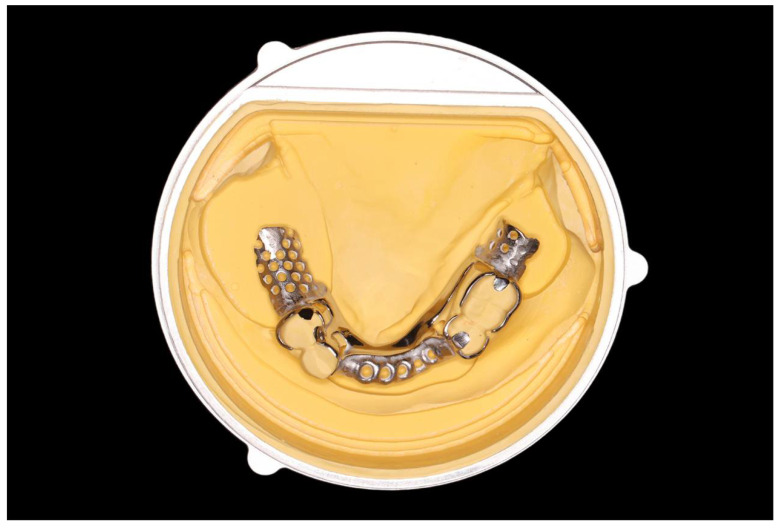
Metal framework seated on the modified cast within a metal jig.

**Figure 5 ijerph-19-08540-f005:**
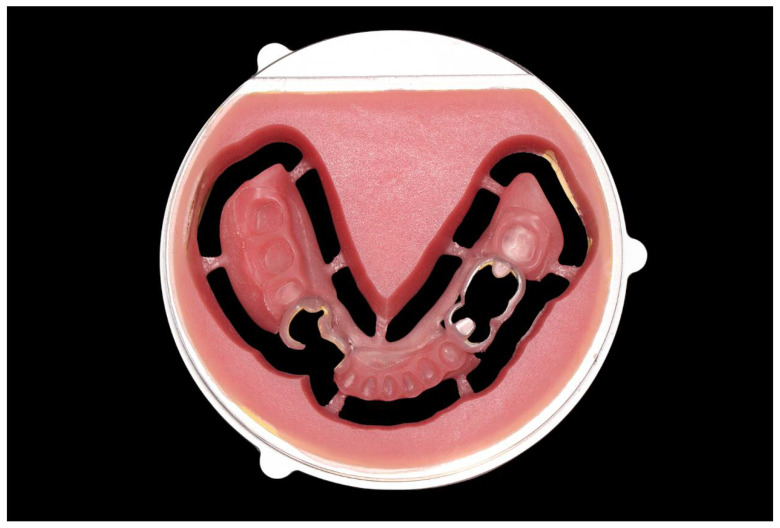
Milled denture base and offset metal structure complex.

**Figure 6 ijerph-19-08540-f006:**
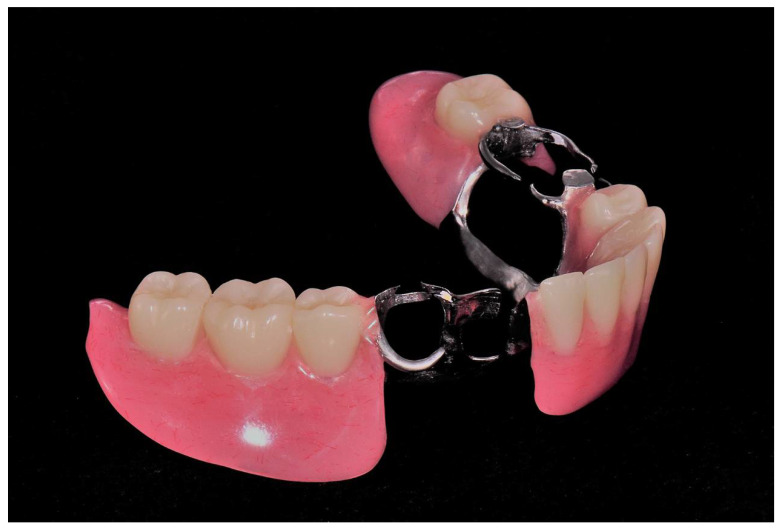
Completed removable partial denture.

## Data Availability

Not applicable.

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
