# Peer review of "A Digital Workflow for the Fabrication of a Milled Removable Partial Denture"

_ijerph, 2022, doi:10.3390/ijerph19148540_

Round 1
Reviewer 2 Report
This is a technical note concerning a very interesting topic in the field of removable prosthodontics which allows dental practitioners to take advantage of the rapid advancements in computer-aided design and computer-aided manufacturing (CAD-CAM) in the fabrication of removable partial dentures leading to more precise results.
The authors describe clearly the steps of their technique and for that they use relevant illustrations that explain very well the work.
However, I would like to highlight on the following:
1. According to the authors, this technique presents only one limitation, whereas in my opinion they can find others such as difficult clinical situations etc.
2. Surely, errors exist when using any technique. What are these errors and how to avoid them.
Reviewer 3 Report
2. As I understand the master casts were scanned, and then the metal framework was designed. Does this mean that the design of the framework was not carried out initially in the position of the centric relation and this position was recorded later at the stage of fitting the already fabricated framework? Why the position of centric relation was not taken into account at the stage of the framework design?
3. Centric relation registration was carried out with a framework fixed in the mouth?
4. “Mill the modified mandibular cast from the gypsum within a metal jig attached to the milling unit (71)” – how does gypsum appear inside the metal jig: are these special gypsum blocks for milling, or the gypsum is poured inside a metal jig? What is the height of the gypsum block?
5. As far as can be judged from the presented drawings and descriptions, the autopolymerizing resin should overlap (be higher) the level of the modified cast within a metal jig. To what depth the gypsum was milled then? How the calculation of the required height of the gypsum block was carried out?
6. What amount of residual monomer is contained in the self-polymerizing resin used in comparison with the resins commonly used for removable dentures?
Reviewer 4 Report
Current manuscript describes a digital workflow for the fabrication of a milled removable partial denture. I think it is so interesting and can be published after major revision.
1. Abstract and conclusion are very short. They must contain obtained data.
2. Introduction is poor. Please rephrase it.
3. Section 2 must be divided into subsections e.g., 2.1. Materials and 2.2. Methods
4. Methods must be explained with details
5. Please discuss about advantages and limitation of this technique
Round 2
Reviewer 3 Report
Thank you for answers! They make the article more clear.
Reviewer 4 Report
Accept in current form.